Hydraulic Redistribution Decreases with Precipitation Magnitude and Frequency in a **Dryland Ecosystem: A Data-Model Fusion Approach** 2 Aneesh Kumar Chandel<sup>1</sup>, Mitra Cattry<sup>3</sup>, Yu Zhou<sup>4</sup>, Hang Duong<sup>2,5</sup>, Marcy Litvak<sup>2</sup>, William 3 Pockman<sup>2</sup> and Yiqi Luo<sup>1</sup> 4 5 <sup>1</sup>School of Integrative Plant Science, Cornell University, Ithaca, NY, USA <sup>2</sup>University of New Mexico, Albuquerque, NM, USA 6 7 <sup>3</sup>Department of Earth and Environmental Engineering, Columbia University, USA 8 <sup>4</sup>Department of Environmental Systems Science, ETH Zurich, Switzerland 9 <sup>5</sup>Vietnam National University of Agriculture, Hanoi, Vietnam 10 Corresponding Authors: Aneesh Kumar Chandel (akc76@cornell.edu), Yiqi Luo 11 (yl2735@cornell.edu) 12 13 14 15 16 17 18 19 20

Abstract

Hydraulic redistribution (HR), the movement of water via plant root systems that connect soil compartments with different water potential, should influences soil moisture dynamics particularly in water-limited ecosystems. Realistic representation of HR in ecosystem models is essential to improve the ability of these models to predict ecosystem function in dryland regions. In this study, we integrated HR into the Terrestrial ECOsystem model and employed a Bayesian Markov Chain Monte Carlo technique to optimize soil hydraulic parameters and root conductance using four years of soil moisture observations from a piñon-juniper woodland. We found that (i) integrating HR generally improved model prediction of soil moisture during dry periods, particularly in the top 30 cm of the soil profile, where more than 50% of root biomass exists, mostly during dry periods; (ii) HR increased surface soil moisture by up to 60% during dry periods; (iii) HR decreased with increasing precipitation magnitude and frequency, however, the length of dry spells between rainfall events also influenced HR rates; and (iv) upward HR in the top 60 cm soil profile became more pronounced as dry conditions progressed, with rates ranging from 0.10 to 0.50 mm d<sup>-1</sup>. These findings highlight that HR plays a likely role in sustaining soil moisture during extended dry periods and has a limited effect during precipitation events. Future research should investigate the effect of HR on other ecosystem processes, such as net ecosystem exchange of carbon and evapotranspiration under varying climatic conditions.

### 1. Introduction

Drylands, cover over 40% of Earth's terrestrial surface and support more than 38% of the 45 global population (Prăvălie, 2016). Ecosystem function in these regions is likely to be limited by 46 altered precipitation in the changing climate (Beer et al., 2010; Ukkola et al., 2021). 47 Understanding the ability of plants to mitigate the potential negative impacts of alter 48 precipitation is therefore critical for predicting ecosystem stability. Hydraulic redistribution (HR) 49 is the passive movement of water through plant roots, usually at night, from wet to dry regions of 50 the plant rooting volume driven by differences in water potential. This passive process can favor 51 plant survival during droughts by tapping into deep soil layers having relatively higher water 52 potential and redistributing water to the shallow root zone (upward HR) (Barron - Gafford et al., 53 2017; Brooks et al., 2002; Dawson, 1993; Domec et al., 2010; Nadezhdina et al., 2015; Nicola et 54 al., 2020; Nicola & Ram, 2022; Prieto et al., 2012). During wet seasons, HR can redistribute 55 water from wet surface soil into deeper, drier soil (downward HR), supplementing the infiltration 56 process in recharging deeper soil layers (Bleby et al., 2010; Fu et al., 2016; Hultine et al., 2003; 57 Scott et al., 2008). Despite its potential role in regulating plant and ecosystem productivity, 58 nutrient cycling and soil microbial activity (Grünzweig et al., 2022; Sardans & Peñuelas, 2014), 59 HR is often ignored in ecosystem models. 60 Hydraulic redistribution has been observed across diverse ecosystems and plant species 61 (Nadezhdina et al., 2010; Neumann & Cardon, 2012; Priyadarshini et al., 2016; Yu et al., 2013), 62 and has been interpreted as structuring dryland plant communities, regulating ecosystem 63 productivity, and enhancing resilience to climate extremes (Barron - Gafford et al., 2021; 64 Barron - Gafford et al., 2017; Hafner et al., 2020; Lee et al., 2018). The dynamics of HR are influenced by various biotic (rooting architecture, plant capacitance, transpiration demand, 65

66 senescence, and dormancy), abiotic factors (soil hydraulic characteristics, soil moisture status), 67 and climatic conditions (precipitation) (Katul & Sigueira, 2010; Prieto et al., 2012; Wei et al., 2022). While several studies have examined HR in deserts and semi-arid ecosystems and 68 69 reported upward HR during dry events (Xing - Ming Hao et al., 2013; Lee et al., 2018; Scott et 70 al., 2008; Yu et al., 2013) and downward HR following precipitation {Hultine, 2003 #2122}, 71 studies focusing on fine-scale temporal variations in HR across different soil depths and multiple 72 years remain limited. Additionally, a quantitative understanding of how precipitation magnitude 73 and frequency influence HR rates, key limiting factors in dryland ecosystems, remain poorly 74 understood. 75 In this study, we explicitly test two hypotheses: (1) Direction of HR: Upward HR should be 76 the dominant form of HR in dryland ecosystem. This is due to the recharge of deeper soil layers 77 from precipitation which can retain moisture for longer periods, and during dry periods roots 78 facilitate the movement of this retained water to the drier surface soils. (2) HR-precipitation 79 relationship: upward HR should decline following precipitation events, reaching its maximum 80 rates during prolonged dry periods as the drought create steep water potential gradients between 81 deeper, moist soil layers and the drier surface layers, facilitating the upward movement of water. 82 Soil moisture dynamics are governed by a complex interplay of forces that drive water 83 movement through the soil profile. The primary drivers include matric potential (capillary and 84 adsorptive forces binding water to soil particles), gravitational potential (driving downward 85 drainage), and potential gradients that induce processes like HR (Caldwell et al., 1998). These 86 forces collectively determine water retention, redistribution, and plant availability (Hillel, 2003). 87 Isolating their individual contributions from field soil moisture data is challenging, as their 88 effects are concurrent and modulated by soil properties, root activity, and atmospheric

90 moisture data, provides a robust framework to isolate and quantify HR, offering a more 91 mechanistic and quantitative understanding. 92 Several modeling studies have incorporated various HR schemes into process-based models 93 to improve understanding of hydrological and ecological processes (Amenu & Kumar, 2008; Fu 94 et al., 2016; Lee et al., 2018; Ouijano & Kumar, 2015; Ryel et al., 2002; Tang et al., 2015; Wu et 95 al., 2020; Zheng & Wang, 2007). However, realistic representation and estimation of parameters 96 related to HR remains a challenge, as neither the magnitude of HR nor its associated parameters 97 can be directly observed in the soil (Quijano & Kumar, 2015; Ryel et al., 2002). As a result, most 98 models rely on default HR parameter values from Ryel et al. (2002) (Fabian et al., 2010; Zheng 99 & Wang, 2007) or estimated parameters using soil moisture data during specific periods of time 100 when upward or downward HR is assumed negligible, such as wet or dry season (Amenu & 101 Kumar, 2008; Fu et al., 2018; Fu et al., 2016; Yan & Dickinson, 2014). The challenges in direct 102 measurements, and reliance on assumed parameter values, constitute key gaps in our 103 understanding of HR dynamics. 104 To address these gaps, we focused on piñon-juniper (PJ) woodlands, the most widespread 105 semiarid ecosystem in the US. PJ woodlands are spatially widespread, ecologically important, 106 temporally dynamic, and structurally unique dryland ecosystem in the western US, spanning 10 107 US states and 40 million hectares across the American Southwest (Eastburn et al., 2024; Romme 108 et al., 2009). Despite their importance, HR has not been previously reported in PJ woodlands. 109 However, our continuous root sap flux measurements provided direct evidence of HR in both 110 piñon and juniper roots, indicated by sustained negative root sap flux during nighttime at the 111 study site (Fig. S.1).

conditions. Consequently, data-model fusion approach, integrating process-based model with soil

In this study, we used the process-based Terrestrial ECOsystem (TECO) model to (i) develop and implement a data assimilation approach to incorporate HR into the TECO model; (ii) quantify and characterize the magnitude and dynamics of HR across multiple soil depths; and (iii) analyze the temporal patterns of HR and its relationship with precipitation magnitude and frequency. The TECO model is a well-established ecosystem model that integrates ecological processes to simulate carbon, water, and energy fluxes within terrestrial ecosystems (Weng & Luo, 2008). We employed data assimilation to constrain the TECO model including HR using four years of soil moisture data measured at multiple soil depths, encompassing both wet and dry periods.

## 2. Data and Methods

# 2.1 Study site and data

Our modeling study utilized data from a PJ woodland plot (Lat. 35.642, Long. -104.607, elevation 1925 m) located in New Mexico, USA, and previously described in Schwinning et al. (2020). The site is a private ranch covering an area of over 6800 hectares that was ungrazed from 2012 through the measurement period used for this study and is characterized by a semi-arid climate. Mean annual precipitation of the site is approximately 460 mm, with the majority falling between May and October, and a mean annual temperature of 10.5 °C. The soil texture at the site varies with depth, ranging from loam to clay loam. The vegetation consists of distinct tree clusters dominated by piñon pine (*Pinus edulis* (Englem.)) and juniper (*Juniperus monosperma* (Englem.) Sarg.) separated by open areas of bare soil and herbaceous cover.

SWC was continuously monitored using multi-sensor frequency domain capacitance probes (Decagon EC-5) installed at four depths (5, 15, 30 and 60 cm), in four soil pits under the tree canopies. All sensors were monitored every minute by a datalogger (model CR6, Campbell

Scientific), storing 15-minute averages were stored by a data logger. For this study, we used the average SWC across all four pits. Each sensor was calibrated in the lab before installation for both air and water frequency. Significant shifts in soil temperature can affect both soil permittivity and the response of capacitance sensors, potentially confounding the small fluctuations in VWC caused by HR. Therefore, temperature correction factors were applied to the measured VWC at each depth, using the nearest measured temperature, following the method described by Saito et al. (2009).

TECO is a process-based ecosystem model (Hou et al., 2021; Jiang et al., 2018; Weng &

## 2.2 Modeling framework

Luo, 2008), and has evolved from the TCS model (Luo & Reynolds, 1999). The model has four 145 major components: canopy photosynthesis, plant growth, soil water dynamics, and soil carbon 146 transfers. The canopy photosynthesis and soil water dynamics submodels run at the hourly time 147 step whereas the plant growth and soil carbon submodels run at the daily time step. The model is 148 driven by seven environmental variables, including precipitation (mm), wind speed (m s<sup>-1</sup>), solar radiation (W m<sup>-2</sup>), air and soil temperature (C), relative humidity (%), and vapor pressure deficit 149 150 (kPa). The detailed description of TECO model is available (Weng & Luo, 2008) and only the 151 brief description of soil water dynamics is provided here. 152 The soil profile is divided into 10 layers, with varying thickness: 5 cm for the first layer, 10, 153 15, and 30 cm for the second, third, and fourth layers respectively, and 20 cm for the fifth to 154 tenth layers. SWC in each layer results from the mass balance between influx and efflux, with 155 changes primarily attributed to vertical unsaturated flow, transpiration, precipitation, runoff, and 156 drainage. Evaporation depletes water from the first two soil layers, while transpiration depletes 157 water from all soil layers containing roots, allocated based on root fraction in each layer (Eq. 8).

- Given the predominantly arid conditions of the study site, runoff and drainage were found
- negligible. Thus, water movement between soil layers is simulated as follows:

$$\frac{dW_i}{dt} = \frac{dF_i}{dz} - E_i - T_i \tag{1}$$

- where  $W_i$  is the water storage (cm) in layer i, t is time (h),  $F_i$  is net unsaturated flow of water into
- layer i (cm h<sup>-1</sup>), z is vertical thickness,  $E_i$  and  $T_i$  are evaporation and transpiration water loss from
- layer i (cm h<sup>-1</sup>).
- The unsaturated soil water movement is simulated vertically according to modified form of
- Buckingham-Darcy's law (Campbell, 1985) (Eq. 2), with Brooks (1965) equation (Eq. 4)
- estimating hydraulic conductivity and soil water retention curve (SWRC) to simulate soil water
- potential  $(\Psi)$ .

$$\frac{dF_i}{dz} = K(\theta_i) \left( \frac{d\Psi_i}{dz} + 1 \right) \tag{2}$$

- where  $K(\theta_i)$  is the unsaturated soil hydraulic conductivity (cm h<sup>-1</sup>) for SWC  $\theta$  (cm<sup>3</sup> cm<sup>-3</sup>) in layer
- i,  $\Psi_i$  is soil water matric potential (MPa) in layer i, and z is the vertical thickness (cm) of the soil.

$$K(\theta_i) = K_s \left[ \frac{\theta_i - \theta_r}{\theta_s - \theta_r} \right]^{(2m+3)}$$
 (3)

- where,  $K_s$  is the soil saturated hydraulic conductivity (cm h<sup>-1</sup>), m is the pore size distribution
- index,  $\theta_s$  and  $\theta_r$  are saturated and residual SWC (cm<sup>3</sup> cm<sup>-3</sup>)

$$\frac{\theta - \theta_r}{\theta_s - \theta_r} = \left(\frac{\Psi}{\Psi_b}\right)^{-1/m} \tag{4}$$

- $\Psi_b$  is the soil air entry water potential.
- To quantify the direction and magnitude of HR, we integrated the HR model by Ryel et al.
- (2002) into equation 1 of TECO model (presented in equation 5). This HR model empirically
- describes HR flux based on the soil water potential gradient between two soil layers (Eq. 6). HR
- was assumed to occur only at night, with its occurrence controlled by solar radiation instead of

fixed day and night hours. Daytime starts as solar radiation exceeds 10 W m<sup>-2</sup>, thereby inhibiting
HR since the water potential gradient typically favors water movement from roots to canopy to
meet transpiration demand during the day. This pattern is evident in Fig. S1, where under low or
zero solar radiation, root sap flux was found to be negative, indicating water movement away
from the root zone which is an indicator of occurrence of HR at the study site. Using these
assumptions, the net water movement into soil layer *i* from other soil layers can be expressed as:

$$\frac{dW_i}{dt} = \frac{dF_i}{dz} - E_i - T_i + H_i \tag{5}$$

$$H_i = C_{RT} \sum (\Psi_j - \Psi_i) max (c_i, c_j) \frac{R_i R_j}{1 - R_r} D_{tran}$$
 (6)

$$c_i = \frac{1}{1 + \left(\frac{\Psi_i}{\Psi_{50}}\right)^b} \tag{7}$$

$$R_i = \frac{R_0}{1 + \left(\frac{d}{d_{50}}\right)^a} \tag{8}$$

Where in Eq 6,  $H_i$  is the net water redistributed by roots into layer i (cm h<sup>-1</sup>),  $C_{RT}$  is the maximum radial soil-root conductance of the entire active root system for water (cm MPa<sup>-1</sup> h<sup>-1</sup>),  $\Psi$  is soil matric potential (MPa),  $c_i$  is a factor reducing soil-root conductance based on  $\Psi_i$ ,  $R_i$  is the fraction of active roots in layer i,  $R_0$  is the average vertically summed root dry mass from the bottom to the root zone to the soil surface, and  $D_{tran}$  is a factor reducing water movement among layers by roots while plant is transpiring and is assumed to be 1.0 during the night when transpiration is minimal and 0.0 during day.  $R_x = R_i$  when  $\theta_i > \theta_j$  or  $R_x = R_j$  when  $\theta_j > \theta_i$ . In Eq 7,  $\Psi_{50}$  is the soil water potential (MPa) where conductance is reduced by 50% and b is an empirical constant. In Eq 8, d is soil depth (cm), and  $d_{50}$  is the soil depth at the median of the root distribution and a is a shape parameter (Table 1). The Brooks (1965) model for SWRC was utilized to simulate soil water potential ( $\Psi$ ), facilitating the development of soil water potential

gradients necessary for HR by tree roots (Eq. 4). Due to lack of site-specific parameters, the default values of b and  $\Psi_{50}$  were used as 3.22 and -1 MPa, respectively (Ryel et al., 2002).

## 2.3 Data assimilation for parameters estimation

We used Bayesian probabilistic inversion to calibrate parameters associated with soil hydraulics, where posterior probability density functions of parameters are obtained from prior knowledge about the parameters and the error between model and observations. According to Mosegaard and Sambridge (2002), Bayesian inversion can be summarized by the following equation:

$$p(c|Z) \propto p(Z|c) p(c) \tag{9}$$

where p(c|Z) is posterior probability density function of model parameters c; p(Z|c) is a likelihood function of parameters c; p(c) is prior probability density function of parameters c. We assumed that the prediction errors were normally distributed and uncorrelated, hence, the likelihood function, p(Z|c), was calculated as follows:

$$p(Z|c) \propto exp\left\{-\sum_{i=1}^{k} \frac{(Z_i - X_i)^2}{2\sigma_i^2}\right\}$$
 (10)

where  $Z_i$  is observed VWC at  $i^{th}$  soil layer,  $X_i$  is VWC simulated by TECO at a corresponding soil depth;  $\sigma_i^2$  is the variance of a measurement at a soil layer; k is the total number of soil layers.

To generate the posterior distributions, we first specified the priors of the parameters to be uniformly distributed over the intervals specified in Table 1. We put constraints on parameters based on the literature. The initial set of parameters was randomly selected within the prior parameter ranges. Once we specified parameter ranges, we used the Metropolis-Hastings (M-H) algorithm (Hastings, 1970; Metropolis et al., 1953), a Markov chain Monte Carlo method, to

sample from the posterior parameter distribution. To generate a parameter set, we ran M-H algorithm in two steps: proposing step and a moving step. In the proposing step, a new parameter set  $c^{new}$  was generated from a previously accepted parameter set  $c^{k-l}$  through a proposal distribution  $(c^{new}|c^{k-l})$ :

$$c^{new} = c^{k-l} + r \times \frac{c^{max} - c^{min}}{D}$$
 (11)

- The value of  $P(c^{k-l}|c^{new})$  was then compared with a random number U from 0 to 1. Parameter set  $c^{new}$  was accepted if  $P(c^{k-l}|c^{new}) \ge U$ , otherwise  $c^k$  was set to  $c^{k-l}$ . In the moving step, a
- probability of acceptance  $P(c^{k-1}|c^{new})$  was calculated as in the following (Marshall et al., 2004):

$$P(c^{k-l}|c^{new}) = min\left\{1, \frac{p(Z|c^{new})p(c^{new})}{p(Z|c^{k-1})p(c^{k-1})}\right\}$$
 (12)

- The M-H algorithm was repeated for 50,000 simulations, and then all accepted parameters values were used to generate the probability distribution functions (Xu et al., 2006). Finally, before each model simulation with optimized parameters, we ran the model for 200 years, a spin-up period that was long enough to obtain stable carbon stock as an initial condition for these simulations.
- To evaluate the impact of HR on soil moisture dynamics in a PJ woodland, we conducted two multi-year simulations using two configurations of the TECO model: TECO+HR (with HR) and default TECO (HR turned off). To distinguish the influence of HR from soil hydraulic properties, we adopted a data assimilation approach focused on calibrating only the TECO+HR model. We calibrated TECO+HR model using soil moisture data measured at 5, 15, 30, and 60 cm depths over a four-year period. The range of prior parameter values was informed by available soil texture data for the study site (Brooks, 1965; Carsel & Parrish, 1988). Within this

- range, we optimized depth-specific soil hydraulic parameters to achieve a close match between
- modeled and observed soil moisture (Table 1).
- After calibrating the TECO+HR model, we deactivated the HR process and ran simulations
- with the same optimized parameters to generate the default TECO scenario. This approach
- allowed us to ensure that differences in soil moisture dynamics between TECO+HR and default
- TECO simulations were attributable solely to the presence or absence of HR. The motivation to
- calibrate only TECO+HR model, rather than the default TECO is to avoid parameter
- compensation for unresolved processes (Luo & Schuur, 2020), in which the absence of HR could
- lead to unrealist adjustments of soil hydraulic parameters to indirectly capture its effects.

# 252 **2.4 Statistical analyses**

- Model performance was assessed by comparing simulated outputs with observed data during
- full simulation periods (2018-2021), dry, and wet periods, defined as days without and with
- rainfall events, respectively. Evaluation was conducted using statistical metrics, including root
- mean square error (RMSE), and absolute mean error (MAE).

$$RMSE = \sqrt{\frac{1}{n} \sum_{i=1}^{n} (m_i - o_i)^2}$$
 (13)

$$MAE = \frac{1}{n} \sum_{i=1}^{n} |m_i - o_i|^2$$
 (14)

- Where:  $o_i$  represents observed values,  $m_i$  represents modeled values, and n represents the
- number of data points.

Table 1: Parameters constrained using data assimilation in the TECO model from soil moisture data from 2018 to 2021.

(Schwinning et al., 2020) (Schwinning et al., 2020) (Schwinning et al., 2020) (Ryel et al., 2002) (Ryel et al., 2002) References Calibrated Calibrated Calibrated Calibrated Calibrated Calibrated cm MPa-1 h-1  $cm^3 cm^{-3}$ cm<sup>-3</sup> cm h-1 kg m<sup>-2</sup> Units MPa cm cm cm [0, 0.08]Range [0, 100][0.1, 2][0, 1][0, 1]Symbols Constrained values 0.05/0.07/0.06/0.03 0.14/0.29/0.30/0.70 0.34/0.38/0.36/0.33 0.89/0.66/0.88/0.84 96/60/50/40 0.022 3.22 -1.0 0.90 2.2 25  $C_{RT}$  $\Psi_{50}$  $D_{50}$  $K_{s}$  $\Psi_b$  $R_0$  $\theta_s$  $\theta_{\dot{r}}$ ш 9 aSoil Ψ where root conductivity reduced by 50% Soil depth at the median of the root distribution Average vertically summed root dry mass Maximum radial soil-root conductance Root distribution shape parameter Saturated hydraulic conductivity Air entry water potential Saturated water content Residual water content Pore size distribution Empirical constant **Parameters** 

Four values represent parameters in the four modeled SWC at depths of 5, 15, 30, and 60 cm, respectively.

3. Results

## 3.1 Parameter estimation via data assimilation and water mass balance

The data assimilation approach, using observational SWC data to constrain the model, yielded well-constrained soil hydraulic parameters (Table 1; Fig. S2 and S3). The resulting posterior probability density functions, characterized by sharp peaks, narrow spread, and consistency across soil depth support the reliability and accuracy of these calibrated parameter values. Additionally, soil water mass balance of soil profile was conserved before and after incorporating the HR process into the TECO model (Fig. S4). The key components of the water budget: precipitation, evapotranspiration, and changes in soil water content remained balanced, ensuring that the model accounted for all water fluxes. Furthermore, the sum of HR across all soil layers (10 layers) was consistently equal to zero, further ensuring that no water was artificially introduced or lost from the system.

## 3.2 Observed and simulated soil moisture

-Observation TECO+HR—default TECO (a) 5cm Volumetric soil moisture (%) Precipitation (mm) (c) 30cm (d) 60cm

Figure 1a-d: Observed and simulated soil volumetric water content for the year 2019 (January 1, 2018 to December 31, 2021) at soil depths of 5 cm (a), 15 cm (b), 30 cm (c), and 60 cm (d). Vertical bars indicate

Figure 2: Model performance for soil moisture across different depths (5, 15, 30, and 60 cm, and 0-60 cm integrated soil profile), considering temporal variations in soil moisture conditions. Root Mean Square Error (RMSE) and Mean Absolute Error (MAE) are presented for the complete time series (a, b), dry periods (c, d), and wet periods (e, f).

The data assimilation-constrained models, generally captured both the magnitude and dynamics of observational data, reproducing seasonal variations in soil moisture across four soil depths. While TECO+HR simulation showed an improvement in the overall model performance, the impact of HR was mostly pronounced during dry periods (Fig. 1 and 2). We further examined diurnal soil moisture fluctuations (Fig. S5) and found that TECO+HR successfully reproduced the observed diurnal cycles, whereas the default TECO failed to capture this pattern, suggesting that the observed diurnal variability was likely driven by HR. Additionally, we compared minmax normalized soil matric potential at 15, 30, and 60 cm with simulations derived from Eq. (4) (Fig. S6). Both models reproduced the general trends of the observations, suggesting that the simulated soil water potential gradients were consistent with measurement.

Moreover, during periods of limited precipitation, the TECO+HR (blue lines) consistently maintained higher soil moisture compared to default TECO (red lines), aligning closer to observation particularly in the topsoil layers (Fig. 1a-c). Following precipitation events, the

311 default TECO and TECO+HR simulations converged, suggesting the minimal influence of HR 312 under wet conditions at the study site. However, as surface soil moisture decreased following 313 precipitation, the two simulations diverged again, with TECO+HR maintaining higher moisture 314 levels in the topsoil layers, highlighting the role of HR in maintaining soil moisture during 315 prolonged drought. 316 The incorporation of HR into TECO resulted in reductions in model errors. During dry 317 periods, the RMSE decreased by 25, 43, and 52% at 5, 15, and 30 cm soil depths, respectively. 318 However, limited improvement was observed at 60 cm soil depth. Correspondingly, the MAE 319 was reduced by 30, 53, and 60% at 5, 15, and 30 cm, respectively. Over the entire study period, 320 RMSE decreased by 24, 25, and 47% at 5, 15, and 30 cm, with MAE reductions were 29, 34, and 321 55% at the same depths (Fig. 2a-d). Overall soil profile performance improved as well, with 322 RMSE and MAE reductions over 40% for both the four-year simulation and dry periods. These 323 improvements during dry periods are especially important, as roots are most vulnerable to 324 drought. By mitigating soil water deficits in surface layers, HR could reduce the risk of hydraulic 325 failure, thereby supporting plant species survival and it could enable better prediction of 326 ecosystem responses to water stress, such as carbon uptake (Domec et al., 2010), and 327 evapotranspiration (Zhu et al., 2017). In contrast, during wet periods, HR had minimal influence 328 on soil moisture (Fig. 2e, f). 329

#### 3.3 HR simulations

Model simulation revealed distinct patterns of HR dynamics across soil depths and temporal scales. Fig. 3 illustrates these patterns over two timescales: a short-term, diurnal pattern (Fig. 3a), and a long-term perspective from 2018 to 2021 (Fig. 3b). HR is a process with both a source and a sink for water movement. In the Fig. 3, positive HR suggest that a soil layer is gaining water (sink), whereas negative HR values suggest that the layer is losing water (source).

Figure 3: Temporal dynamics of hydraulic redistribution (HR). (a) diurnal pattern of modeled hydraulic redistribution across soil depths from July 22-24, 2018. The graph illustrates HR patterns during a dry period followed by a precipitation event. Colored lines represent different soil depths. (b) long-term daily HR trends and precipitation from January 2018 to December 2021. The blue shaded area represents precipitation (right y-axis).

The short-term modeling analysis

highlights diurnal pattern of HR during dry conditions and a precipitation event (Fig. 3a). For instance, on July 23, 2018, during a dry period, upward HR occurred, moving water from deeper (> 100 cm) to shallower (0-30 cm) soil layers. However, following a precipitation event on July 24, 2018 (12 mm), this pattern shifted. The top 5 and 15 cm layers showed negative HR and a decrease in the upward HR rate, respectively, acting as a water source for deeper layers. At the same time, deeper soil layers showed a decline in negative HR rates, suggesting signs of receiving water likely from the topsoil layers. The sum of HR across all soil layers remained

zero, confirming that HR redistributed water rather than adding to the system. Consequently, downward HR from the topsoil supplemented infiltration, enhancing water movement into deeper soil layers, reflected by a decrease in the negative HR rates at depths and an increase in the positive HR rate at 30 cm (Fig. 3a). While our model simulates HR across 10 soil layers, we present long-term results for only the top four soil layers (5, 15, 30, and 60 cm) to enable direct comparison with the available observed soil moisture data. A clear seasonal pattern emerged, with HR generally intensifying during dry periods (Fig. 3b). Our model showed that upward HR was predominantly occurring in up to top 30 cm of soil profile, with values ranging from -0.066 to 0.29 mm d<sup>-1</sup> in each soil layer and an average of 0.30 mm d<sup>-1</sup> across the top 30 throughout the study period. Downward HR, while less pronounced, moved water only from the 5 cm soil layer during monsoon seasons and large precipitation events (e.g., July 2018, 2019, 2020, and 2021; Fig. 3b). In contrast, 60 cm soil layer typically exhibits a negative HR during dry periods, acting as a water source for upper layers, and positive HR during wet periods, suggesting occasional water input from surface layers ranging from -0.096 to 0.059 mm d<sup>-1</sup> (mean 0.0015 mm d<sup>-1</sup>). Moreover, integrated soil profile (top 60 cm of soil profile), showed that upward HR was the dominant form of HR throughout the year, ranging from 0.10 to 0.53 mm d<sup>-1</sup> with a mean value 0.31 mm d<sup>-1</sup> (Fig. S7).

## 3.4 Precipitation influences on HR

Figure 4: Relationships between HR and precipitation. (a) Weekly HR rates versus weekly precipitation amounts at different soil depths (5, 15, 30, and 60 cm). Trend lines and  $R^2$  values are shown for each depth. (b) Weekly HR (0–60 cm) versus weekly precipitation, with trend line and  $R^2$  value.

The model results showed a significant linear relationship between weekly HR and precipitation (mm week $^{-1}$ ) (Fig. 4a-b). In the topsoil layers (5 cm and 15 cm), negative correlations were observed ( $R^2 = 0.40$  and 0.28 respectively, both p-values 

consecutive rainfall events lengthen (Fig. 5). HR was lowest under conditions of high rainfall frequency and shorter dry spells, progressively increasing to its peak in the absence of rainfall. However, as the drought period extended beyond 30 days, HR declined, suggesting potential limitation on availability of deeper water to sustain HR. This variability is further illustrated through three scenarios (Fig. S7): 1) Following a

Figure 5: Relationship between weekly mean hydraulic redistribution (mm d<sup>-1</sup>), dry spell length between two rainfall events (log<sub>10</sub>(days+1)). The color scale indicates the number of rainfall events per week, while marker size represents the weekly precipitation amount (mm week<sup>-1</sup>). Dry spell length denotes the number of rainless days between two consecutive precipitation events.

rainfall event (28 mm on July 5, 2018), HR in the top 60 cm of soil profile was minimal at 0.13 mm d<sup>-1</sup>, indicating limited driving force for water redistribution when soil moisture was abundant. 2) During a transition period between rainfall events (July 5-10, 2018), HR gradually increased but remained moderate, ranging from 0.13 to 0.20 mm d<sup>-1</sup>, suggesting a progressive activation of the redistribution process as soil began to dry. 3) During a prolonged dry period (November 23-30, 2018), HR peaked at 0.20-0.52 mm d<sup>-1</sup>, demonstrating enhanced redistribution activity in response to the development of soil moisture gradients.

## 4. Discussion

# 4.1 Patterns of hydraulic redistribution

Our findings support the hypothesis that upward HR is the dominant form of HR in dryland ecosystems due to limited precipitation amount and sporadic rainfall events (Fig. S7). This prevalence of upward water movement is characteristic of semi-arid regions, where deep-rooted plants often redistribute water from moist deeper layers to drier surface soils during periods of

414 water stress (Caldwell et al., 1998; Ryel et al., 2002). Notably, the most pronounced HR 415 occurred in the topsoil layer (5, 15, and 30 cm), (Fig. 3b), which can be attributed to vertical root 416 distribution, with over 50% of root biomass concentrated in the top 30 cm ( $D_{50} = 25$  cm) of the 417 soil profile (Fig. S8). This pattern aligns with findings of Xing Ming Hao et al. (2013), which 418 suggest that deeper root distributions extend HR to deeper soil layers, while shallower root 419 systems enhance HR in the topsoil layers due to higher root density and activity (Fig. S8). 420 Our model simulations estimated HR rates in range 0.10-0.53 mm d<sup>-1</sup> for top 60 cm soil depth (Fig. S7), values that fall within the broader range of 0.04 to 3.2 mm d<sup>-1</sup> reported in the 421 422 comprehensive review by Neumann and Cardon (2012) but exceed the upper limit of the 95% confidence interval of 0.014-0.475 mm d<sup>-1</sup> reported by Yang et al. (2022) for desert or sparsely 423 424 vegetated ecosystems, which synthesized empirical observations and modeling estimates of 425 average water movement attributed to HR. 426 The direct impact of HR on hydrological processes should be evident in the soil profile water 427 content. We tested this by comparing SWC model simulations in TECO with and without HR 428 processes, to observed SWC time series at four depths (Fig. 6). We found that cumulative effects 429 of HR on soil moisture vary with depth, primarily due to the non-uniform root biomass 430 distribution throughout the soil profile (Fig. S8). The most pronounced effects of HR were 431 observed in the topsoil layers (5, 15, and 30 cm), where average daily water content increased by 432 up to 60% compared to simulation without HR. This increase was driven by upward HR, 433 especially during dry-down periods (Fig. 3b, and 6). 434

22

Figure 6: Relative change in soil water content (SWC) (%) compared to observed SWC at 5, 15, 30, and 60 cm depths. The color gradient represents the magnitude of relative change in SWC, calculated as (HR–No HR)/No HR×100, with HR and No HR indicating simulations with and without hydraulic redistribution."

# 4.2 Effects of precipitation variability on HR

Our findings support the hypothesis that the precipitation pattern significantly (p-value < 0.001) influences the magnitude and variability of HR (Figs. 3a-b, and S7). The rate of HR in the topsoil profile (<60 cm) exhibited a consistent response pattern to precipitation events, characterized by sharp decreases following large rainfall, and gradual recovery to pre-rain levels. These dynamics were particularly evident during years with frequent large precipitation events (2018-2019), where HR rates oscillated between 0.04- and 0.20-mm d<sup>-1</sup>. For instance, after an 18 mm rainfall event on July 24, 2018, HR rates dropped below 0.15 mm d<sup>-1</sup> before recovering to 0.30 mm d<sup>-1</sup> within 10 days. This pattern suggests a recharging effect: initially, infiltrating rainwater increases soil water potential in both shallow and deep layers, reducing the gradient between shallow and deep layers and temporarily suppressing HR (Xing Ming Hao et al., 2013).

However, as water redistributes through the soil profile, new hydraulic gradients develop, 448 leading to enhanced HR activity. In this phase, roots actively redistribute water from newly 449 moistened deep layers to drier shallow layers, consistent with findings from Yu and D'Odorico 450 (2014) and (Ryel et al., 2002). 451 Our model predicted that HR rates were generally higher during rainless periods compared to 452 rainfall periods within a given year. For instance, during the prolonged dry period in 2020 (driest 453 year with few small precipitation events), HR rates remained consistently high, 0.17-0.40 mm d 454 1, with minimal fluctuations. The consistent high HR rates, likely arises from more pronounced 455 soil water potential gradients derived from sustained plant water demand and surface evaporation 456 in the absence of frequent precipitation (Fu et al., 2016; Meinzer et al., 2004). 457 Seasonal trends include higher HR rates (0.12-0.53 mm d<sup>-1</sup>) during the drier periods (typically November to May) and lower rates (0.10-0.30 mm d<sup>-1</sup>) during the monsoon season 458 459 (usually June to October) (Fig. S7). This seasonality underscores the influence of both 460 precipitation patterns and potential evapotranspiration on HR dynamics, highlighting that HR is 461 likely more pronounced during drier seasons when soil moisture gradients are likely to be more 462 substantial due to reduced precipitation and potentially higher evaporative demand (Fu et al., 463 2016; Scott et al., 2008; Yu & D'Odorico, 2014). 464 4.3 Limitation and future perspectives 465 While our modeling study provides valuable insights into HR dynamics in PJ woodlands, 466 several limitations should be noted: (1) The model does not account for inter-annual changes in 467 vegetation cover or species composition. Variations in plant functional types and leaf area index 468 may influence soil moisture and HR, and incorporating these dynamics could improve long-term 469 simulations. (2) We focused on dominant tree species at our study site, but other plants may

benefit from water redistributed by these trees, affecting ecosystem water dynamics. (3) We did
not include stem water refilling or nighttime transpiration reported by Howard et al. (2009);
Neumann et al. (2014) which could influence the magnitude of HR. (4) Finally, future studies
could explore the role of HR in regulating ecosystem functions, such as carbon exchange and
evapotranspiration, and examine whether incorporating HR improves predictions of ecosystem
carbon dynamics proportionally to its effects on soil water.

## 5. Conclusions

This study demonstrates the role of hydraulic redistribution (HR) in soil water dynamics in piñon-juniper woodlands. By integrating HR processes and observations into the Terrestrial Ecosystem Model (TECO) via data assimilation, we successfully constrained model soil hydraulics parameters and improved simulations of soil water content across multiple depths, particularly in shallow soil layers (0–30 cm) and during dry periods. Our model results showed that HR rates vary with the length of dry spells between rainfall events, generally decreasing with increasing precipitation magnitude and frequency, with HR rates ranging from 0.10 to 0.50 mm d<sup>-1</sup> as conditions transitioned from wet to dry. Consequently, HR increased soil moisture in topsoil layers by up to 60% during dry periods, with upward HR emerging as the dominant flux, especially in the top 30 cm. These findings underscore the potential influence of HR during dry periods and highlight its role in sustaining soil water availability for vegetation. Future research should explore how HR-mediated water redistribution affects ecosystem functions including carbon exchange, and evapotranspiration.

6. Acknowledgments This research is supported by US Department of Energy's Office of Biological and 491 492 Environmental Research, Environmental System Science (ESS) Program (DE-SC0023514 to 493 WTP, MEL and YL) and the modeling effort used data collected with funding from the US 494 National Science Foundation (IOS 1557176 to MEL, WTP, and Susan Schwinning). Additional 495 support from US NSF grants (DEB 2242034, DEB 2406930, and DEB 2425290), the US 496 Department of Energy's Terrestrial Ecosystem Sciences Grant DE-SC0023514, subcontract 497 CW55561 from Oak Ridge National Laboratory to Cornell University, and the "NYS Connects: 498 Climate Smart Farms & Forestry" project, funded by the USDA, the New York State Department 499 of Environmental Conservation, and the New York State Department of Agriculture and 500 Markets. This research is also part of AI-CLIMATE: "AI Institute for Climate-Land Interactions, 501 Mitigation, Adaptation, Tradeoffs and Economy", funded by the USDA National Institute of 502 Food and Agriculture (NIFA), the NSF National AI Research Institutes Competitive Award (No. 2023-67021-39829) and Swiss National Foundation, Award#P500PN 206603 for their financial 503 504 support of this collaboration. 505 7. Competing interests 506 The authors declare no competing interests. 507 8. Author contributions 508 AKC: Conceptualization, Methodology, Data curation, Formal analysis, Writing - original draft. 509 YZ: Conceptualization, Writing - review & editing. MC: Writing - review & editing. HD: 510 Writing - review & editing, Data curation. ML: Conceptualization, Data curation, Supervision, 511 Writing - review & editing, Funding acquisition, Project administration. WP: Conceptualization, 512 Data curation, Supervision, Funding acquisition, Writing - review & editing. YL:

- 513 Conceptualization, Supervision, Project administration, Funding acquisition, Writing review &
- editing.

# 515 9. Code availability

- The Terrestrial ECOsystem (TECO) model used in this paper is available on GitHub at
- <a href="https://github.com/aneeshchandel/TECO">https://github.com/aneeshchandel/TECO</a> HR.

# 518 **10. Data availability**

- The data supporting the findings of this study are available within the manuscript. Additional
- data may be available upon request from the corresponding author, subject to compliance with
- relevant data protection and privacy regulations.

# 522 11. Supporting Information

- Supporting information accompanying this manuscript is available as a separate Word file. It
- includes supplementary figures referenced in the main text.

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
