# Peer review of "Hydraulic Redistribution Decreases with Precipitation Magnitude and Frequency in a Dryland Ecosystem: A Data-Model Fusion Approach 2 Aneesh Kumar Chandel1, Mitra Cattry3, Yu Zhou4, Hang Duong2,5, Marcy Litvak2, William 3 Pockman2"

_EGUsphere, 2025_

## Author Comment (AC1)

**Reviewer #1**

(Manuscript number: egusphere-2025-4608)

We sincerely thank reviewer 1 for the thoughtful evaluation and for recognizing the novelty and timeliness of our work on incorporating hydraulic redistribution into a terrestrial ecosystem model. We also thank the reviewer for the constructive suggestions aimed at improving the presentation and emphasizing the broader implications of the work. We have carefully considered each recommendation and have revised the manuscript accordingly to enhance clarity, coherence, and impact. Reviewer comments are shown in blue italic, followed by our detailed responses. We hope these revisions address all concerns satisfactorily.

*The manuscript "Hydraulic Redistribution Decreases with Precipitation Magnitude and Frequency in a Dryland Ecosystem: A Data-Model Fusion Approach" tested the incorporation of algorithms that represent hydraulic redistribution in a terrestrial ecosystem model for improved estimates of hydrological processes, which are also used to inform mechanistic understanding of hydraulic redistribution in a dryland ecosystem under different climate conditions. This is a timely and novel contribution because limited mechanistic understanding and limited modeling of hydraulic redistribution can lead to large uncertainty in estimates of hydrological processes. However, the presentation of the work can be improved to better illustrate the importance and implications of the study, and I have the following suggestions for consideration:*

Response: We greatly appreciate the reviewer for the constructive assessment of our study.

*1. Consider starting the introduction with soil moisture, since most readers are more familiar with that topic. Introduce matric potential and other narrower concepts before discussing hydraulic redistribution and model details. The current introduction does not flow from broad to narrow and it uses terms such as water potential before those terms receive a clear definition later on.*

Response: The introduction has been revised to improve the flow from broad concepts (soil moisture) to more specific processes (water potential and hydraulic redistribution), with all technical terms now defined prior to use.

*2. Consider providing more justification for model improvements and give a more detailed review of existing models and their gaps to place this study in the current literature. Connect the detailed hypotheses more clearly to the modeling activities.*

Response: The justification for model development has been strengthened with a detailed review of existing modeling approaches and their limitations, and the study hypotheses are now more clearly linked to the modeling objectives.

*3. State quality control procedures for the soil moisture data, and give the exact depth and time step of the soil moisture records used for parameterization. Those details are currently unclear in the methods and can only be inferred from the results.*

Response: The Methods section now includes a detailed description of soil moisture data quality control procedures, as well as explicit information on the depth and temporal resolution of the records used for model parameterization, as follows:

"Soil volumetric water content (VWC) was continuously monitored using multi-sensor frequency domain capacitance probes (Decagon EC-5) installed at four depths (5, 15, 30 and 60 cm), in four soil pits under the tree canopies. All sensors were monitored every minute by a datalogger (model CR6, Campbell Scientific), and 15-minute averages were stored. For model parameterization, we used 15-min VWC records aggregated to daily means. Each sensor was calibrated in the lab before installation for both air and water frequency. Because soil temperature can affect both soil permittivity and the response of capacitance sensors, potentially confounding the small fluctuations in VWC caused by HR, temperature correction factors were applied to the measured VWC at each depth, using the nearest measured temperature, following the method described by Saito et al. (2009). Rather than excluding data below 0 °C, we used this temperature-correction approach to reduce the influence of temperature-driven artifacts on the soil moisture signal. This strategy allows retention of continuous soil moisture records while accounting for the known sensitivity of capacitance sensors to temperature-dependent changes in dielectric permittivity."

*4. Consider using an antecedent precipitation index or a similar metric to represent cumulative precipitation effects, because precipitation often produces lagged responses in hydrological processes.*

Response: We thank the reviewer for this insightful suggestion. We agree that cumulative precipitation and lagged hydrological responses are important for understanding soil moisture dynamics and HR. In response, we have incorporated an antecedent precipitation index (API) to represent precipitation memory effects in our analysis.

Specifically, we added the following description to the Methods (section 2.3):

"To evaluate the influence of cumulative precipitation and soil moisture memory on HR, we calculated the Antecedent Precipitation Index (API) for the study period (2018–2021) following Kohler and Linsley (1951). API acts as a proxy for soil moisture status by accounting for the decaying effect of past rainfall events. The daily API ($API_t$) was calculated using the recursive decay function:

$$API_t = P_t + (k*API_{t-1}) \qquad (13)$$

where $P_t$ is the precipitation on day $t$ (mm), $API_{t-1}$ is the index value of the preceding day, and $k$ is a decay constant representing the recession of soil moisture due to evapotranspiration and drainage. We used a decay constant of $k = 0.90$, which falls within the commonly applied range for antecedent precipitation indices and is consistent with optimization analyses indicating

optimal decay constants near 0.90 (Li et al., 2021). This metric enables differentiation between short dry intervals following wet conditions and extended dry spells with limited antecedent moisture."

We have updated Results (section 3.5) as follows:

[Figure]

*Figure 5: Relationships between hydraulic redistribution (HR) and Antecedent Precipitation Index (API). (a) Weekly HR rates versus mean weekly API at different soil depths (5, 15, 30, and 60 cm). For each depth, the trend lines, R², and corresponding p-value are shown. (b) Depth-integrated weekly HR across 0–60 cm soil profile versus mean weekly API, with trend line, R², and p-value.*

*5. Consider reducing repetition between the discussion and the results. Also consider comparing this study with similar work, expanding mechanistic explanations where needed, and discussing future implications more thoroughly to strengthen the discussion section.*

Response: We have revised the Discussion to reduce overlap with the Results, added comparisons with related studies, expanded the mechanistic interpretation of our findings, and strengthened the discussion of broader implications and future directions to improve the overall depth and clarity of the section.

**Specific comments:**

*L21: "Dryland" should be emphasized in the abstract.*

Response: We have revised the opening sentence of the abstract to explicitly emphasize dryland ecosystem as follows:

"Hydraulic redistribution (HR), the movement of water via plant root systems that connect soil compartments with different water potential, should influences soil moisture dynamics particularly in dryland ecosystems, where water availability strongly constrains ecosystem function."

Response: Thank you for the comment. The grouped citations were included to reflect the broad body of literature demonstrating hydraulic redistribution across ecosystems and species. However, we agree that excessive clustering may reduce readability. To improve clarity, we have streamlined the citations while still acknowledging the key foundational studies. The revised sentence now includes a more focused set of representative references rather than an extended list.

*L59: Try to be more specific about how often HR is ignored in models. Also consider discussing this later in the introduction to make the logic flow better.*

Response: Revised as follows:

"Despite its potential role in regulating plant and ecosystem productivity, nutrient cycling and soil microbial activity (Grünzweig et al., 2022; Sardans and Peñuelas, 2014), most current dynamic global vegetation models and Earth system model still lack an explicit representation of HR (Fu et al., 2016)."

*L83. Water potential was used previously before being introduced and defined here. See my suggestions about rearrangement.*

Response: Following the suggested reorganization, we have revised the introduction to ensure that soil moisture is introduced first, followed by water potential.

*L141. Is there quality control to remove data from the freezing point?*

Response: Yes, quality control procedures were applied to address potential temperature effects on soil moisture measurements, including conditions near the freezing point. To improve clarity on soil moisture data quality control, the following text has been added to the Methods section:

"Soil volumetric water content (VWC) was continuously monitored using multi-sensor frequency domain capacitance probes (Decagon EC-5) installed at four depths (5, 15, 30 and 60 cm), in four soil pits under the tree canopies. All sensors were monitored every minute by a datalogger (model CR6, Campbell Scientific), and 15-minute averages were stored. For model parameterization, we used 15-min VWC records aggregated to daily means. Each sensor was calibrated in the lab before installation for both air and water frequency. Because soil temperature can affect both soil permittivity and the response of capacitance sensors, potentially confounding the small fluctuations in VWC caused by HR, temperature correction factors were applied to the measured VWC at each depth, using the nearest measured temperature, following the method described by Saito et al. (2009). Rather than excluding data below 0 °C, we used this temperature-correction approach to reduce the influence of temperature-driven artifacts on the soil moisture signal. This strategy allows retention of continuous soil moisture records while accounting for the known sensitivity of capacitance sensors to temperature-dependent changes in dielectric permittivity."

*Table 1: What sources are used to inform the priors for the first six parameters?*

Response: The prior ranges for the soil hydraulic parameters were informed by Rawls et al. (1982) and Clapp and Hornberger (1978) and the HR related parameter ($C_{RT}$) by Fu et al. (2016). We have revised the text as follows:

"The prior range of soil hydraulic parameters were informed by established relationships between soil texture and hydraulic properties (Rawls et al., 1982; Clapp and Hornberger, 1978). The prior range for $C_{RT}$ was based on values reported in Fu et al. (2016)."

*L271: Is the "balanced results" expected because of the method used? If so is this more for methods than results?*

Response: Yes, conservation of soil water mass balance is an expected property of the model formulation, as the TECO model explicitly enforces water conservation through its mass-balance equations. This framework is described in the Methods section, where we detail how the soil profile is divided into 10 layers (to 180 cm depth) and how volumetric water content in each layer is updated based on the balance between incoming and outgoing fluxes, including vertical unsaturated flow, evapotranspiration, precipitation, runoff, and drainage. Evaporation is restricted to the upper two layers, while transpiration is distributed across rooting layers according to the prescribed root fraction. Under the predominantly arid conditions of the study site, runoff and drainage are negligible, and water movement among layers is governed primarily by internal redistribution processes.

Although mass balance conservation is therefore a methodological expectation, we reported it in the Results section to demonstrate that incorporating hydraulic redistribution (HR) did not introduce numerical artifacts or spurious gains or losses of water. Verifying that soil water mass balance remains conserved after adding HR serves as an important diagnostic outcome of the simulations, confirming that the HR implementation is physically consistent with the original TECO model structure.

*Figure 2: State how many days or percentage of days are in the "dry" and "wet" periods, respectively. If the data size is very different they may influence the interpretation of error metrics.*

Response: We agree that clarifying the relative sample sizes improves the interpretation of error metrics. We found that wet days account for 22% of the total simulation periods, while the remaining 78% correspond to dry days. We have added this information to the text associated with Figure 2 as follows:

"Model performance was assessed by comparing simulated outputs with observed data during full simulation periods (2018- 2021), dry, and wet periods, defined as days without and with rainfall events, respectively. During the study period, wet days accounted for 22% of all days, whereas dry days comprised the remaining 78%."

*L426. This paragraph seems more like results than discussion.*

Response: The suggested paragraph has been moved to the results under a new section 3.3 Effects of HR on soil moisture.

***Technical comments:***

*L44: Remove the comma after "drylands".*

Response: Fixed.

*L70. Fix the broken citation.*

Response: The broken citation has been fixed.

*L132. Spell out abbreviations at their first appearances.*

Response: We have revised the manuscript to spell out all abbreviations at their first occurrence.

*L135. "Storing" and "store" seem repetitive.*

Response: The sentence has been updated as "All sensors were monitored every minute by a datalogger (model CR6, Campbell Scientific), and 15-minute averages were stored."

*L417 and L446. The citation does not seem correct since it has the full name of an author.*

Response: Thanks for pointing it out. The citation format has been corrected.

**Reference**

Clapp, R. B. and Hornberger, G. M.: Empirical equations for some soil hydraulic properties, Water resources research, 14, 601-604, 1978.

Fu, C. S., Wang, G. L., Goulden, M. L., Scott, R. L., Bible, K., and Cardon, Z. G.: Combined measurement and modeling of the hydrological impact of hydraulic redistribution using CLM4.5 at eight AmeriFlux sites, Hydrology and Earth System Sciences, 20, 2001-2018, 10.5194/hess-20-2001-2016, 2016.

Grünzweig, J. M., De Boeck, H. J., Rey, A., Santos, M. J., Adam, O., Bahn, M., Belnap, J., Deckmyn, G., Dekker, S. C., and Flores, O.: Dryland mechanisms could widely control ecosystem functioning in a drier and warmer world, Nature ecology & evolution, 6, 1064-1076, 2022.

Kohler, M. A. and Linsley, R. K.: Predicting the runoff from storm rainfall, US Department of Commerce, Weather Bureau1951.

Li, X., Wei, Y., and Li, F.: Optimality of antecedent precipitation index and its application, Journal of Hydrology, 595, 126027, 2021.

Rawls, W. J., Brakensiek, D. L., and Saxtonn, K.: Estimation of soil water properties, Transactions of the ASAE, 25, 1316-1320, 1982.

Saito, T., Fujimaki, H., Yasuda, H., and Inoue, M.: Empirical temperature calibration of capacitance probes to measure soil water, Soil Science Society of America Journal, 73, 1931-1937, 2009.

Sardans, J. and Peñuelas, J.: Hydraulic redistribution by plants and nutrient stoichiometry: Shifts under global change, Ecohydrology, 7, 1-20, 2014.